# Public Services in the Household and Their Effect on Poverty, Analysis for the Peruvian Case, 2021

**Santotomas Licimaco Aguilar-Pinto** [1], **Julio Cesar Quispe-Mamani** [2,*], **Dominga Asunción Calcina-Álvarez** [3], **Nelly Jacqueline Ulloa-Gallardo** [4], **Roxana Madueño-Portilla** [4], **Mindi Fabiola Lizárraga-Álvarez** [5], **Adderly Mamani-Flores** [6], **Balbina Esperanza Cutipa-Quilca** [7], **Ruth Nancy Tairo-Huamán** [8], **Duverly Joao Incacutipa-Limachi** [6] and **Marleny Quispe-Layme** [9]

1. Faculty of Administrative Sciences, Andean University Nestor Caceres Velasquez, Taparachi Urbanization Km 4.5 Exit to Puno, Juliaca 322213, Peru; d02291995@uancv.edu.pe
2. Faculty of Economic Engineering, National University of the Altiplano, Floral Avenue 1153, Puno 21001, Peru
3. Academic Department of Education and Humanities, Faculty of Education, Amazon National University of Madre of Dios, Puerto Maldonado 17001, Peru; dcalcina@unamad.edu.pe
4. Faculty of Engineering, Amazon National University of Madre of Dios, Puerto Maldonado 17001, Peru; nulloa@unamad.edu.pe (N.J.U.-G.); rmadueno@unamad.edu.pe (R.M.-P.)
5. Department of Engineering of the Industry and the Environment, San Pablo Catholic University, Campiña Paisajista Urbanization, Arequipa 04001, Peru; mflizarraga@ucsp.edu.pe
6. Faculty of Social Sciences, National University of the Altiplano, Floral Avenue 1153, Puno 21001, Peru; adderlymamani@unap.edu.pe (A.M.-F.)
7. Faculty of Accounting and Administrative Sciences, National University of the Altiplano, Floral Avenue 1153, Puno 21001, Peru; becutipa@unap.edu.pe
8. Basic Sciences Academic Department, Amazon National University of Madre of Dios, Puerto Maldonado 17001, Peru
9. Ecotourism Faculty, Amazon National University of Madre of Dios, Puerto Maldonado 17001, Peru
* Correspondence: jcquispe@unap.edu.pe

**Abstract:** The objective of the research was to determine the effect of public services in the household on poverty in Peru, in the period 2021, for which a quantitative, non-experimental research approach was considered with a descriptive and correlational design. The information from the National Household Survey of the National Institute of Statistics and Informatics (INEI) database was used, considering the modules "Dwelling and Household Characteristics", "Household Members' Characteristics", "Education", "Employment and Income", "Household Equipment", "Summaries (Calculated Variables)" and "Citizen Participation". It was possible to determine that the following variables had negative effects on household poverty in Peru: access to potable water, sanitation, electric power, cell phone services; achieving higher, secondary, and primary education levels; having a washing machine, motorcycle, tricycle, motorcycle taxi, computer, kitchen, refrigerator in the household; having a property title; being part of an association or organization; living in a rural residence area; and having remittances. However, the number of household members had a positive effect on poverty. Therefore, it was concluded that access to public services in the household contributed to reducing the probability of being poor in Peru.

**Keywords:** public services; household; poverty; social inequality; public policies

## 1. Introduction

In general, until today, the existence of different types of development is evident. The developed and positioned countries form a smaller group, while intermediate countries are in the process of consolidation towards development, and there are also groups of countries that are poor and have unequal development, where a population with an economic income that is below the world average can be identified. There are also low levels of productivity, deficiency in household and housing services (potable water service, sanitation, electricity,

cell phone, complementary household assets and services, physical and legal sanitation conditions of the property, area of residence, and reception of remittances, among others), and mainly low quality of life for members. This inequality occurs due to the way in which economic, social, and environmental policies are implemented, among other reasons. Being part of a political, economic and social system, policies follow the path established by the country, but they are sometimes the least adequate for being able to contribute towards the development of the country (Bergara and Andrés 2005; Olarte 2017; Velayarce et al. 2022; Bramley and Watkins 2008; Birner and Von Braun 2015; Hewett and Montgomery 2001; Aaberge et al. 2010; Booysen et al. 2008; Bardhan 2006; Fan et al. 2000; Griggs et al. 2013; Castillo and Huarancca 2022; Csapo 2023; Barbier 2022; Ruijer et al. 2022; Abrar ul Haq et al. 2022; Guo et al. 2023).

At the global level, objectives have been established at different meetings carried out by different countries and world organizations, such as the World Bank Group, to end extreme poverty and at the same time guarantee the promotion of shared prosperity. There is a commitment to meet these goals by more than 140 countries; however, as of 2023, there are still people who survive in extreme poverty with less than 2.15 dollars/day. The decrease in world poverty was permanent until before the pandemic, but this was interrupted by the COVID-19 crisis, causing loss of jobs, including temporary or permanent layoffs. Military conflicts and the effects of climate change have also affected the development of programs, projects, and public policies aimed at reducing poverty, which was evidenced through the decrease in household economic income (Comisión Económica para América Latina y el Caribe 2021; Cepal 2022; Yang et al. 2020; Wodon et al. 2006; Nguyen and Nordman 2018; Leng et al. 2021; UNICEF 2000; Banco Mundial 2020; Hou et al. 2023; World Bank Group 2022; World Bank Group 2023; Akyüz 2017). Therefore, despite the fact that poverty in the world has resumed its downward trajectory, in 2022, an average of 75–95 million people lived in extreme poverty (Griggs et al. 2013; Ludwig et al. 2013; Aramburú and Rodríguez 2011).

Additionally, as of 2019, billions of people worldwide lacked access to potable water and sanitation services; 2200 million people did not have access to water services in a timely and safe manner; 4200 million people in the world were not part of the management for access to sanitation services; and the most critical thing was that 3000 million people did not have any type of basic facilities for hand washing and complementary hygiene. Additionally, only 1.8 billion people had access to basic service of potable water since 2000, of which 1 in 10 people did not have basic services in their household. This gap had widened considerably by 2023, thus showing the existence of inequality in access, availability, and quality of basic services. The aforementioned is more critical in rural areas, since of that group of people who did not have access to basic services, 8 out of 10 people who lived in rural areas did not have access to basic services (Comisión Económica para América Latina y el Caribe 2021; Cepal 2022; Yang et al. 2020; Wodon et al. 2006; Nguyen and Nordman 2018; Leng et al. 2021; UNICEF 2000; Banco Mundial 2020; Hou et al. 2023; World Bank Group 2022; World Bank Group 2023; Akyüz 2017; Abrar ul haq et al. 2019; Abrar ul haq et al. 2018).

In the case of Latin America and the Caribbean, the existing gap in household public services is not unrelated to what is happening in the world, given that 17 million people still do not have access to electricity services and 75 million people do not have access to fuel and clean technology to prepare and cook their food. This makes these populations more vulnerable to poverty, which was worsened during the pandemic, since they were affected on the one hand by the increase in the price of fossil fuel and on the other hand by the health crisis that occurred between 2020 and 2022. This generated energy insecurity that directly affected the physical, social, and economic conditions of millions of people. In addition, according to the data reported by ECLAC, on average 15% of the population living in precarious housing conditions do not have access to electricity. In the case of Bolivia, Chile, El Salvador, Honduras, Guatemala, and Nicaragua, this indicator is between 30% and 40%, which coincides with the living conditions in households, as these are mostly

found in informal settlements where the electricity service is of poor quality, there is poor infrastructure and households have electrical appliances that are deficient, in poor condition, or deteriorated (Moshinsky 1959; Bruno Besana et al. 2015; Homedes and Ugalde 2002; Wiesenfeld and Sánchez 2012; Yaschine 2015; Jordán and Simioni 2003; Monterrosa 2023; Marina Clemente et al. 2018; Galilea and Antúnez 2003; Barrantes 2005; Insulza 2014; United Nations 2020; Comisión Económica para América Latina y el Caribe 2021; Organización de las Naciones Unidas para la Alimentación y la Agricultura 2020).

Regarding access to potable water service, 161 million people do not have adequate access to potable water service; that is, 3 out of 10 people do not have optimal potable water service. The case of sanitation services is more critical, given that 431 million people do not have adequate access; that is, 7 out of 10 people do not have optimal sanitation services. This phenomenon occurs due to different aspects, such as the existence of little spending being assigned and executed at the levels of different governments in turn, given that spending on potable water in households only reaches 0.8% of the total. Because of the little impact of potable water and sanitation projects and programs on the quality of life of families and the existence of deficient and inadequate services, households seek to satisfy these needs through other options, such as the purchase of bottled water or tanker trucks. The households in the lower economic income quintile pay up to 3 times more than those in better conditions; for example, the cost of water per tanker truck in Cochabamba-Bolivia is 4 times higher than the cost of piped water, while in Peru this cost is 12 times higher (Moshinsky 1959; Bruno Besana et al. 2015; Homedes and Ugalde 2002; Wiesenfeld and Sánchez 2012; Yaschine 2015; Jordán and Simioni 2003; Monterrosa 2023; Marina Clemente et al. 2018; Galilea and Antúnez 2003; Barrantes 2005; Insulza 2014; United Nations 2020; Comisión Económica para América Latina y el Caribe 2021; Organización de las Naciones Unidas para la Alimentación y la Agricultura 2020; Álvarez-Risco et al. 2020; Castillo and Huarancca 2022).

From this international and Latin American perspective, Peru is a state that is not immune to poverty and in which there was a marked increase in the scarcity of resources during the 1980s and early 1990s. Impoverishment continues to be a prominent issue for the government, and over the years attempts have been made to reduce it (Calatayud Mendoza 2020). During these last years, economic growth was fighting poverty in the country, until the arrival of the pandemic, during which the level of poverty increased due to the paralysis of economic activities. The event of extreme poverty in 2020 affected 5.1% of the population, representing 1 million 664 thousand people, and this population had a level of per capita spending lower than the cost of the basic food hamper. Compared to the previous year, poverty increased from 2.9% in 2019 to 5.1% in 2020, increasing by 2.2% of the population; in other words, in 2020 there were 732,000 more people living in extreme poverty than in 2019. On the other hand, the regions that witnessed more poverty in the 2020 period were: Ayacucho, Cajamarca, Huancavelica, Huánuco, Pasco, and Puno (INEI 2021; Barillas 2010).

In addition, in analyzing the importance of public services in Peru, as of 2011 (as a scenario without a pandemic), only 82.50% of households had access to potable water, sanitation, and electricity services, showing a greater gap in access to sanitation in urban and rural areas, which only covered 75.2% of households. The result was reflected in high rates of existing diseases such as anemia and gastrointestinal and diarrheal diseases. However, by 2021, these indicators had improved, since households with access to water service through the public network increased to 90.60%. In the case of sanitation or sanitary sewer service, this access increased by 8%, since it was 68.20% in 2013 and 76.30% in 2021. In addition, with respect to the electric power service, access to said service increased by 11.60%, since it was 82.50% in 2011 and it reached 94.10% in 2021. Complementarily, the municipal service of sweeping urban streets increased from 64.40% in 2020 to 71.20% in 2021, and the municipal service of household garbage collection improved from 97.40% in 2020 to 98.20% in 2021 (Herrera 2002; Sánchez 2015; INEI 2019, 2011, 2021; Mendoza et al. 2010; Vásquez Huamán 2012; Álvarez-Risco et al. 2020; Morán-Mariños et al. 2019; Instituto Nacional

de Estadística e Informática INEI 2022; Ministerio de Desarrollo e Inclusión Social 2021; Castillo and Huarancca 2022).

According to the indicators shown in the previous paragraphs, improving access to public services in households is optimal but there is a gap to be closed by geographical area. In the mountains and jungle, these services are still not efficient and, in many cases, do not meet the expectations of the users, since despite the fact that these basic services are regulated by the State, they are not related in a symbiotic way between economic and social dimensions. By not optimally covering access to these services in households, there is a direct relationship between the lack of access to services and affordability problems. In addition, access to public services positively influences the improvement of people's quality of life, allowing a reduction in the poverty level and the vulnerability of the population to being poor and extremely poor, thereby guaranteeing equal opportunities (Valenzuela 2013; Durán 2013; Parra 2011; Barneche et al. 2010; Castillo and Huarancca 2022; INEI and BM Banco Mundial 2000; Quispe-Mamani et al. 2022). Therefore, according to the above information, the objective of the research was to determine the effects of public services in the household on poverty in Peru, in the period 2021.

## 2. Literature Review

### 2.1. General Aspects of Poverty

Poverty cannot be defined exactly since there is controversy over its definition by different authors and institutions. According to Caloca et al. (2017), based on the theories of Ricardo and Sen, poverty is defined as the non-existence of freedom, equality, equity, and non-violence in all its meanings, in the opportunities for women and men to fulfill themselves, both in the private and public spheres, and in their interactions with their environment and with other individuals. The ability to achieve their well-being will be diminished by these factors to such a degree that it will bring with it the concrete absolute and relative deprivation of satisfaction of basic needs and interests to have a flourishing life, and without further ado, this group of women and men will be poor. This definition reveals the economic, social, and emotional inequality that different people face.

On the other hand, poverty is the deprivation of well-being in a pronounced way, that is, the lack of access to basic capacities to function in society and an adequate income to meet the needs of education, health, security, empowerment, and basic rights (Galindo and Ríos 2015). Additionally, Spicker (2009) defines poverty as the inability to achieve a minimum standard of living, where said amount varies from one country to another and reflects the cost of participating in the daily life of societies.

For the Banco Mundial (2005), poverty is defined as the inability to achieve a minimum standard of living, which is why a level is established based on consumption that consists of two elements: the necessary expense to access a minimum standard of nutrition and other very basic needs, and an amount that varies from one country to another that reflects the cost of participating in the daily life of societies. The first element is calculated by looking at the prices of the foods that constitute the diets of the poor. The second element is difficult to measure since the installation of drainage in a household is a luxury in some countries and a necessity in others.

According to Song (2017), the main definitions of poverty used in Peru are: total poverty, which is when people in households have a per capita income or consumption less than the cost of a total hamper of assets; extreme poverty, when people in households have a per capita income or consumption less than the value of a minimum food hamper; the poverty gap; which is the average difference between the income of the poor and the value of the poverty lines; severity of poverty, which is an indicator of inequality among the poor; and finally, the population with unsatisfied basic needs, which refers to those who have at least one unsatisfied basic need.

## 2.2. Types of Poverty

### 2.2.1. Monetary Poverty

To measure monetary poverty, the welfare expenditure indicator is used, which is composed of: purchases, self-consumption, self-supply, payments in kind, transfers from other households, and public donations. The people in households whose per capita expenditure is not enough to buy a basic hamper of food and non-food are considered to be monetarily poor. The extreme poor are those who reside in households whose per capita expenditures are below the cost of the basic food hamper (Vásquez Huamán 2012; Vera 2020; León Mendoza 2019).

According to INEI and BM Banco Mundial (2000), the monetary approach is used to measure the incidence of poverty. According to this notion of poverty, all persons residing in private households are considered poor if their per capita expenditure, monetarily valued, does not exceed the poverty line threshold.

The authors Huerta Camones and Milla Aranda (2020), Minaya Aguirre (2021), and Bogale et al. (2005), indicated that three indices are applied to make a measurement of monetary poverty. The first indicates the incidence of poverty, which represents the proportion of poor or extremely poor as a percentage of the total population; that is, it determines the proportion of the population whose consumption is below the value of the poverty line or the value of the extreme poverty line, as the case may be. This measure of poverty does not take into account the magnitude of the gap that separates the spending of the poor from the poverty line, nor does it consider the way in which spending is distributed among the poor.

For this reason, it is complemented with the measurements of the Poverty Gap Index, which measures the average inadequacy of the consumption of the poor with respect to the poverty line, taking into account the proportion of the poor population in the total population and the severity of poverty, which measures inequality among the poor. Based on the aforementioned aspects, these authors developed the following expression, which makes it possible to quantitatively and qualitatively determine the change in the level of poverty of a population (Ariza and Retajac 2020; Colca and Huarancca 2019).

$$FGT_\alpha = \frac{1}{n}\sum_{t=1}^{q}\left(\frac{z - y_t}{z}\right)^\alpha$$

where *FGT* is the Foster, Greer, and Thorbecke Index, *z* is the poverty line, *n* is the number of people in an economy, *q* represents the number of poor (those with expenses equal to or less than *z*), *yi* corresponds to individual expenditures, and $\alpha$ is a sensitivity parameter.

### 2.2.2. Multidimensional Poverty

Because of the limitation that focuses on monetary poverty, multidimensional poverty arises. While one only evaluates the spending of families in a determined period of time and is compared with the poverty line, the other focuses on the poverty index, which is based on indicators that are directly related to education, standard of living, and health, the same indicators that aim to find the deprivations that individuals have or are submerged in the short- and long-term. One of the characteristics of multidimensional poverty is that it allows us to graphically see the deficiencies or needs that monetary poverty, for example, hides or does not reveal; therefore, we can definitively say that it is an instrument that allows us to focus public and social spending (Ariza and Retajac 2020; Reyes and Peguero 2017). In this understanding, we can say that poverty is related to the lack of certain basic elements that prevent the individual from having a full life (Santillán et al. 2020; Dirksen and Alkire 2021).

In 1997, the United Nations Development Program (UNDP) already conceptualized poverty from a multidimensional perspective, as it considered poverty as an impediment to enjoying or living a full life. In simple terms, poverty was the inability to have decent food, the deprivation of a household where one can live, not being able to be in good health, not being able to have an education and, on top of that, not being able to have access to a

decent life. We must also specify that apart from all this, one must also count the security of the individual, the freedom to free thought and ideology, and in addition to that, the associated respect for human rights, the support of stable, productive, and well-paid work, and free development within their community (Pnud 2000; Marina Clemente et al. 2018).

*2.3. Poverty Measurement Methods*

The poverty line method (LP), which is expressed in per capita terms where the poverty line is compared with the income of the household and also per person. Households with incomes below this line are considered poor and those with incomes below the extreme poverty or indigence line are classified as extremely poor (Vera 2005; Feres and Mancero 2001).

The sectoral approach to unmet basic needs (ESNBI), which defines a minimum in each need and calculates the low population of each (illiteracy, water, nutrition, etc.), and leads to fragmentary lists of specific gaps (Laos 2001).

The unsatisfied basic needs method (NBI), which simultaneously considers different dimensions of well-being in households, so it allows the identification of households and poor people (Feres and Mancero 2001).

The method of unsatisfied basic needs with restricted variant, which reduces the analyzed needs to housing and its services, as well as school attendance in minors (Boltvinik 2000).

The method of unsatisfied basic needs with generalized variant, which considers the total of existing basic needs (Díaz 2016).

The integrated poverty measurement method (MMIP), which consists of an integration of NBI and LP methods, such as health and safety care (Reyes and López 2016).

*2.4. Public Services and Poverty*

The implementation of an adequate and efficient policy can guarantee achieving sustained economic growth by seeking to reduce the gap of social and mainly economic inequality among the population, improving economic income and at the same time contributing to the improvement of the conditions of a country for the promotion of public and private investment, and improving the provision of services in the household. In order to guarantee the improvement of the quality of life conditions of household members, it is necessary to guarantee access and conditions of basic services such as potable water, sanitation, electricity, gas, health, education, transportation, and telecommunication, among others (Ellis and Mdoe 2003; Fan et al. 2000; Bapna 2012; Ali and Abdulai 2010; Leng et al. 2021; Zaman and Khilji 2013; Chaskin 2013; Strier et al. 2021; Chang et al. 2020).

Reviewing the theoretical aspect of the relationship between the infrastructure of services and poverty indicated that this topic was little studied. Even when reviewing bibliographic references of the relationship between public services in the household and its relationship with poverty, there was not much research. Therefore, it is necessary to contribute and demonstrate related research on this currently important topic since guaranteeing full and adequate access to services in the household contributes towards the best conditions for improving the quality of life of household members (Bertot et al. 2016; Li and He 2021; Bhagwati 1988; Kakwani and Son 2022; Bruno Besana et al. 2015; Lustig et al. 2012; Kresalja 2017; Kohanoff 2011; Buitrago Betancur and Valencia Agudelo 2007; Barrutia et al. 2022; Ellis and Mdoe 2003).

**3. Materials and Methods**

*3.1. Approach, Design and Type of Research*

The research corresponds to a quantitative, non-experimental approach, with a descriptive and correlational design (Mendoza Bellido 2014; Hernández et al. 2014; Batthyány et al. 2011; Gómez-Peresmitréz and Reidl 2010).

### 3.2. Data Source, Population and Study Sample

Information from the National Household Survey (ENAHO) of the National Institute of Statistics and Informatics (INEI) database was considered using the modules "Dwelling and Household Characteristics", "Household Members Characteristics", "Education", "Employment and Income", "Household Equipment", "Summaries (Calculated Variables)", and "Citizen Participation". The study population considered was the set of all people who were members of the household, who were classified by type of resident in urban and rural areas, and the socioeconomic conditions in which they lived, at the level of all of Peru.

According to the INEI, the sample considered for the ENAHO survey for 2021 was probabilistic, area stratified, multistage, and independent in each study region at the level of Peru, where a confidence level of the sample results of 95% was considered.

To determine the size of the sample at the level of Peru, household members aged 14 years and over were considered, with defined socioeconomic characteristics and their highest level of education attained. The total sample for the study group was 32,199 observations.

### 3.3. Analysis of Variables

The variables considered were dependent and independent. Poverty was measured considering the poverty line established by the INEI, the value of which considered the expenses of a family as a base, since it quantified the standard of living based on what "people and households buy, acquire and consume". In this sense, the person was considered poor when their monthly spending was less than 378 soles, and the "poverty line" was equivalent to the cost of a basic hamper of food and non-food consumption. Therefore, the group of households that had a monthly expense greater than said amount was considered as non-poor and the group with less than that was considered as poor. In the case of independent variables (Table 1).

**Table 1.** Operationalization of variables.

| Variables | Factor | Indicator | Category | Data Type | Source |
|---|---|---|---|---|---|
| Dependent | | | | | |
| Poverty | Economic–social | Poverty line | 1 = Poor | Qualitative | |
| | | | 0 = No poor | | |
| Independent | | | | | |
| Potable water service | Social | Household service | 1 = Has access | Qualitative | National Household Survey (ENAHO)—Survey of living conditions and poverty, 2021 |
| | | | 0 = No access | | |
| Sanitation service | Social | | 1 = Has access | Qualitative | |
| | | | 0 = No access | | |
| Electric energy service | Social | | 1 = Has access | Qualitative | |
| | | | 0 = No access | | |
| Cell phone service | Social | Individual service | 1 = Has access | Qualitative | |
| | | | 0 = No access | | |
| Higher education level | Social | Education Service | 1 = Level reached | Qualitative | |
| | | | 0 = Did not reach the level | | |
| Secondary education level | Social | | 1 = Level reached | Qualitative | |
| | | | 0 = Did not reach the level | | |
| Primary education level | Social | | 1 = Level reached | Qualitative | |
| | | | 0 = Did not reach the level | | |
| household members | Social | Number of household members | Numerical 1 = Yes, received remittances | Quantitative | |

**Table 1.** *Cont.*

| Variables | Factor | Indicator | Category | Data Type | Source |
|---|---|---|---|---|---|
| Has remittance | Económico | Remittance | 0 = Received no remittances | Qualitative | |
| | | | 1 = Yes counts | | |
| Has a washing machine | Social | | 0 = Does not count | Qualitative | |
| | | | 1 = Yes counts | | |
| Has a motorcycle | Social | | 0 = Does not count | Qualitative | |
| | | | 1 = Yes counts | | |
| Count a tricycle | Social | | 0 = Does not count | Qualitative | |
| | | | 1 = Yes counts | | |
| Has a motorcycle taxi | Social | Complementary assets and services in the household | 0 = Does not count | Qualitative | |
| | | | 1 = Yes counts | | |
| Has a computer | Social | | 0 = Does not count | Qualitative | |
| | | | 1 = Yes counts | | |
| Has a kitchen | Social | | 0 = Does not count | Qualitative | |
| | | | 1 = Yes counts | | |
| Has a refrigerator | Social | | 0 = Does not count | Qualitative | |
| | | | 1 = Yes, it has a property title | | |
| Has property title | Social | | 0 = Does not have property title | Qualitative | |
| | | | 1 = Yes, it is part | | |
| Is part of an association | Social | Household belonging, zone and association | 0 = Not part | Qualitative | |
| | | | 1 = Rural area | | |
| Rural residence area | Social | | 0 = Urban area | Qualitative | |
| | | | 1 = Has access | | |

### 3.4. Approach to the Econometric Model

A binary model of the logit-binomial type was considered, and estimations were performed using the maximum likelihood method. The logit-binomial model considered the probability of being poor $P$ (Poor = 1) as the dependent variable; therefore, we sought to determine how public services in the household had an effect on it, according to the following:

$$P(Poverty = 1) = 1/(1 + e(-(\beta_0 + \beta_1 X\_1 + \beta_2 X\_2 + \cdots + \beta_k X\_k + e_i)))$$

Detailing as follows:

$$
\begin{aligned}
P(Poverty = 1) \quad &= 1/(1 + e(-(\beta_0 + \beta_1 Water\ service + \beta_2 Sanitation\ service \\
&+ \beta_3 Electric\ energy\ service + \beta_4 Cell\ phone\ service + \beta_5 Higher\ education\ level \\
&+ \beta_6 Secondary\ education\ level + \beta_7 Primary\ education\ level + \beta_8 Household\ members \\
&+ \beta_9 Has\ remittance + \beta_{10} Has\ a\ washing\ machine + \beta_{11} Count\ as\ motorcycle \\
&+ \beta_{12} Count\ as\ tricycle + \beta_{13} Count\ as\ motorcycle\ taxi + \beta_{14} Count\ as\ computer \\
&+ \beta_{15} Has\ a\ kitchen + \beta_{16} Has\ a\ refrigerator + \beta_{17} Has\ a\ property\ title \\
&+ \beta_{18} Is\ part\ of\ an\ association + \beta_{19} Rural\ residence\ zone + e_i)))
\end{aligned}
$$

It should be emphasized that poverty was measured considering the poverty line established by the INEI, in which the monthly expense was less than 378 soles and the "poverty line" was equivalent to the cost of a basic hamper of food and non-food consumption. A group of households that had a monthly expense higher than said amount was considered as non-poor and one with less than that was considered as poor.

### *3.5. Techniques*

The technique used for data collection and processing was of a documentary nature (books, scientific journals, and other digital materials), where the literature related to the subject was reviewed. Then, the INEI website was accessed, specifically the ENAHO. Subsequently, the selected variables were corrected, data processing and results were obtained using STATA software, and finally, the regression of the proposed model was calculated, descriptively analyzing and relating these variables.

### 4. Results

According to the indicators of monetary poverty in Peru in 2021, poverty reached 25.9% of the total population, which decreased by 4.20% compared to 2020, but increased by 5.70% compared to 2019. All of this was due to the effects of COVID-19, where thanks to the implementation of social policies for social confinement and the economic crisis that occurred at the country and world levels, negative effects were felt not only on the quality of life of households but also on their economic conditions. This was reflected in the data compiled by the National Institute of Statistics and Informatics (INEI), where based on the National Household Survey (ENAHO), poverty reached an average of 19.63% of households in 2021, with a standard deviation of 0.1963 (Table 2). In addition, the poverty line established for 2021 by the INEI was 378 soles/month/person, where if it was less than said amount, the person was considered poor, and if it was less than 201 soles/month/person, the person was classified as extremely poor.

Of the total number of those considered in the ENAHO survey, which was 32,199 people, 81.91% were considered non-poor and 18.09% were considered poor, and this was complemented by the characteristics of poverty by geographical area; in the rural sierra poverty reached 44.30%, in the rural jungle it reached 35.00%, in the urban highland's poverty reached 23.30%, and poverty in the urban jungle was 21.60%. The issue was very differentiated with respect to the rural coast, which reached a poverty level of 21.50%, while in the urban coast it was 17.90% (Table 2).

**Table 2.** Descriptive statistics on public services and poverty.

| Variable | Mean | Standard Deviation | Minimum Value | Maximum Value |
|---|---|---|---|---|
| Poverty | 0.1963 | 0.3972 | 0 | 1 |
| Water service | 0.8280 | 0.3774 | 0 | 1 |
| Sanitation service | 0.6642 | 0.4723 | 0 | 1 |
| Electric energy service | 0.9552 | 0.2068 | 0 | 1 |
| Cell phone service | 0.9453 | 0.2273 | 0 | 1 |
| Higher education level | 0.1662 | 0.3722 | 0 | 1 |
| Secondary education level | 0.5115 | 0.4999 | 0 | 1 |
| Primary education level | 0.7809 | 0.4136 | 0 | 1 |
| Has a washing machine | 0.2974 | 0.4571 | 0 | 1 |
| Has a motorcycle | 0.1113 | 0.3145 | 0 | 1 |
| Count a tricycle | 0.0122 | 0.1096 | 0 | 1 |
| Has a motorcycle taxi | 0.0682 | 0.2521 | 0 | 1 |
| Has a computer | 0.3394 | 0.4735 | 0 | 1 |
| Has a kitchen | 0.8971 | 0.3038 | 0 | 1 |
| Has a refrigerator | 0.5505 | 0.4974 | 0 | 1 |
| Has a property title | 0.3848 | 0.4865 | 0 | 1 |
| Is part of an association | 0.2187 | 0.4134 | 0 | 1 |
| Household members | 3.3574 | 1.7456 | 1 | 15 |
| Rural residence zone | 0.2224 | 0.4158 | 0 | 1 |
| Has remittance | 0.0897 | 0.2858 | 0 | 1 |

Analyzing the behavior of public services in the household, a well-defined behavior could be clearly seen in the case of basic household services. In the case of potable water service, on average 82.80% had access to potable water service (Table 2); in addition, 18.67%

of the people indicated that they did not have access to potable water service and 81.33% had access to potable water service. In the case of the relationship with poverty, the population in a non-poor condition who did not have access to potable water service was 16.89%, while the population in a non-poor condition who had access to potable water service was 83.11%. On the contrary, the population living in poverty without access to potable water service was 26.72%, while the population living in poverty and having access to potable water service was 73.28% (Table 3).

Regarding sanitation service, on average 66.42% had access to sanitation service (Table 2); in addition, 39.90% of the people indicated that they did not have access to sanitation service and 60.10% had access to sanitation service. In analyzing the relationship with poverty, the population in a non-poor condition who did not have sanitation service was 35.27%, while the population in a non-poor condition who had sanitation service was 64.73%. On the contrary, the population living in poverty without sanitation service was 60.90% and the population living in poverty and having sanitation service was 39.10% (Table 3).

In the case of electric power service, on average 95.52% had access to electric power service (Table 2); in addition, 7.14% of the people indicated that they did not have access to electric power service and 92.86% had access to electric power service. In the case of the relationship with poverty, the population in a non-poor condition without electric power service was 5.55%, while the population in a non-poor condition with electric power service was 94.45%. On the contrary, the population in a poor condition without electric power service was 14.34% and the population in a poor condition with electric power service was 85.66% (Table 3).

**Table 3.** Relationship of basic household services with poverty.

| Item | Type of Service | Value | Poverty | | Total |
|---|---|---|---|---|---|
| | | | Not | Yes | |
| Basic household services | Potable water service | Not | 4456 | 1556 | 6012 |
| | | % | 16.89% | 26.72% | 18.67% |
| | | Yes | 21,919 | 4268 | 26,187 |
| | | % | 83.11% | 73.28% | 81.33% |
| | | Total | 26,375 | 5824 | 32,199 |
| | | % | 81.91% | 18.09% | 100.00% |
| | Sanitation service | Not | 9302 | 3547 | 12,849 |
| | | % | 35.27% | 60.90% | 39.90% |
| | | Yes | 17,073 | 2277 | 19,350 |
| | | % | 64.73% | 39.10% | 60.10% |
| | | Total | 26,375 | 5824 | 32,199 |
| | | % | 81.91% | 18.09% | 100.00% |
| | Electric power service | Not | 1464 | 835 | 2299 |
| | | % | 5.55% | 14.34% | 7.14% |
| | | Yes | 24,911 | 4989 | 29,900 |
| | | % | 94.45% | 85.66% | 92.86% |
| | | Total | 26,375 | 5824 | 32,199 |
| | | % | 81.91% | 18.09% | 100.00% |
| | Cell phone service | Not | 1584 | 665 | 2249 |
| | | % | 6.01% | 11.42% | 6.98% |
| | | Yes | 24,791 | 5159 | 29,950 |
| | | % | 93.99% | 88.58% | 93.02% |
| | | Total | 26,375 | 5824 | 32,199 |
| | | % | 81.91% | 18.09% | 100.00% |

In relation to cell phone service, on average 94.53% had access to cell phone service (Table 2); 6.98% of the people indicated that they did not have access to cell phone service and 93.02% had access to cell phone service. In relation to poverty, the population in a

non-poor condition who did not have cell phone service was 6.01%, while the population in a non-poor condition who had cell phone service was 93.99%. On the contrary, the population in a poor condition without cell phone service was 11.42% and the population in a poor condition with cell phone service was 88.58% (Table 3).

Analyzing the education service, a higher level of education was reached on average by 16.62% of respondents, a secondary education level was reached on average by 51.15%, and a primary education level was reached by 78.09% (Table 2). When analyzing the relationship between education and poverty, the populations in a non-poor condition who did not reach the level of higher, secondary, and primary education were 79.50%, 47.07%, and 21.40%, respectively. The populations in a non-poor condition who reached the level of higher, secondary, and primary education were 20.50%, 52.93% and 78.60%. On the contrary, the populations in a poor condition who did not reach the level of higher, secondary, or primary education were 96.07%, 70.24%, and 34.56%; and the populations in a poor condition who completed higher, secondary, and primary school were 3.93%, 29.76%, and 65.44%, respectively (Table 4).

**Table 4.** Relationship of household member characteristics with poverty.

| Item | Type of Service | Poverty | | | Total |
| | | Value | Not | Yes | |
|---|---|---|---|---|---|
| Members, education, and remittances from household members | Higher level of education | Not | 20,969 | 5595 | 26,564 |
| | | % | 79.50% | 96.07% | 82.50% |
| | | Yes | 5406 | 229 | 5635 |
| | | % | 20.50% | 3.93% | 17.50% |
| | | Total | 26,375 | 5824 | 32,199 |
| | | % | 81.91% | 18.09% | 100.00% |
| | Secondary education level | Not | 12,415 | 4091 | 16,506 |
| | | % | 47.07% | 70.24% | 51.26% |
| | | Yes | 13,960 | 1733 | 15,693 |
| | | % | 52.93% | 29.76% | 48.74% |
| | | Total | 26,375 | 5824 | 32,199 |
| | | % | 81.91% | 18.09% | 100.00% |
| | Primary education level | Not | 5644 | 2013 | 7657 |
| | | % | 21.40% | 34.56% | 23.78% |
| | | Yes | 20,731 | 3811 | 24,542 |
| | | % | 78.60% | 65.44% | 76.22% |
| | | Total | 26,375 | 5824 | 32,199 |
| | | % | 81.91% | 18.09% | 100.00% |
| | Household members | 1 | 4868 | 283 | 5151 |
| | | 2 | 6068 | 816 | 6884 |
| | | 3 | 5482 | 928 | 6410 |
| | | 4 | 5111 | 1334 | 6445 |
| | | 5 | 2839 | 1142 | 3981 |
| | | 6 | 1185 | 680 | 1865 |
| | | 7 | 464 | 313 | 777 |
| | | 8 | 200 | 167 | 367 |
| | | 9 | 95 | 80 | 175 |
| | | 10 | 33 | 42 | 75 |
| | | 11 | 12 | 24 | 36 |
| | | 12 | 9 | 10 | 19 |
| | | 13 | 7 | 4 | 11 |
| | | 14 | 1 | 1 | 2 |
| | | 15 | 1 | 0 | 1 |
| | | Total | 26,375 | 5824 | 32,199 |
| | | % | 81.91% | 18.09% | 100.00% |
| | Has remittance | Not | 23,734 | 5474 | 29,208 |
| | | % | 89.99% | 93.99% | 90.71% |
| | | Yes | 2641 | 350 | 2991 |
| | | % | 10.01% | 6.01% | 9.29% |
| | | Total | 26,375 | 5824 | 32,199 |
| | | % | 81.91% | 18.09% | 100.00% |

The number of household members is a variable that directly influences poverty, which is why when studying it, the average number of members per household was 3 people (Table 2). The highest concentration of members per household occurred between 1 person to 5 people, reaching 89.66% (Table 4). In the case of remittances, on average the members of the household indicated that they did not receive remittances to supplement their economic income (Table 2). In addition, 90.71% of the people indicated that they did not receive remittances and 9.29% affirmed that they received remittances. In the case of the relationship with poverty, 89.99% of the people in a non-poor condition did not receive remitances, while 10.01% of the people in a non-poor condition did receive remittances. On the contrary, 93.99% people in a poor condition did not receive remittances and only 6.01% of the people in a poor condition did receive remittances (Table 4).

When analyzing the complementary assets and services of the household, in the case of having a washing machine, motorcycle, tricycle, motorcycle taxi, computer, kitchen, and refrigerator or freezer as complementary assets, it could be seen that an average of 29.74%, 11.13%, 1.22%, 6.82%, 33.94%, 89.71%, and 55.05% of persons had said assets (Table 2). In addition, in relation to poverty, the populations who were not poor and did not have said complementary household assets were 69.74%, 84.86%, 99.01%, 91.92%, 63.06%, 10.25%, and 43.86%, while those in a non-poor condition who had complementary assets in the household were 30.26%, 15.14%, 0.99%, 8.08%, 36.94%, 89.75%, and 56.14%, respectively. Meanwhile the populations in a poor condition who did not have complementary assets in the household were 93.96%, 88.86%, 99.04%, 94.14%, 90.38%, 28.23%, and 78.28%, and those living in poverty and having complementary assets in the household were 6.04%, 11.14%, 0.96%, 5.86%, 9.62%, 71.77%, and 21.72%, respectively (Table 5).

**Table 5.** Relationship of complementary household assets and services with poverty.

| Item | Type of Service | Poverty | | | Total |
| | | Value | Not | Yes | |
|------|-----------------|-------|-----|-----|-------|
| Complementary assets and services in the household | Has a washing machine | Not | 18,395 | 5472 | 23,867 |
| | | % | 69.74% | 93.96% | 74.12% |
| | | Yes | 7980 | 352 | 8332 |
| | | % | 30.26% | 6.04% | 25.88% |
| | | Total | 26,375 | 5824 | 32,199 |
| | | % | 81.91% | 18.09% | 100.00% |
| | Has a motorcycle | Not | 22,382 | 5175 | 27,557 |
| | | % | 84.86% | 88.86% | 85.58% |
| | | Yes | 3993 | 649 | 4642 |
| | | % | 15.14% | 11.14% | 14.42% |
| | | Total | 26,375 | 5824 | 32,199 |
| | | % | 81.91% | 18.09% | 100.00% |
| | Has a tricycle | Not | 26,113 | 5768 | 31,881 |
| | | % | 99.01% | 99.04% | 99.01% |
| | | Yes | 262 | 56 | 318 |
| | | % | 0.99% | 0.96% | 0.99% |
| | | Total | 26,375 | 5824 | 32,199 |
| | | % | 81.91% | 18.09% | 100.00% |
| | Has a motorcycle taxi | Not | 24,244 | 5483 | 29,727 |
| | | % | 91.92% | 94.14% | 92.32% |
| | | Yes | 2131 | 341 | 2472 |
| | | % | 8.08% | 5.86% | 7.68% |
| | | Total | 26,375 | 5824 | 32,199 |
| | | % | 81.91% | 18.09% | 100.00% |

**Table 5.** *Cont.*

| Item | Type of Service | Poverty | | | Total |
| --- | --- | --- | --- | --- | --- |
| | | Value | Not | Yes | |
| | Has a computer | No | 16,633 | 5264 | 21,897 |
| | | % | 63.06% | 90.38% | 68.01% |
| | | Sí | 9742 | 560 | 10,302 |
| | | % | 36.94% | 9.62% | 31.99% |
| | | Total | 26,375 | 5824 | 32,199 |
| | | % | 81.91% | 18.09% | 100.00% |
| | Features a kitchen | Not | 2703 | 1644 | 4347 |
| | | % | 10.25% | 28.23% | 13.50% |
| | | Yes | 23,672 | 4180 | 27,852 |
| | | % | 89.75% | 71.77% | 86.50% |
| | | Total | 26,375 | 5824 | 32,199 |
| | | % | 81.91% | 18.09% | 100.00% |
| | Has a refrigerator | Not | 11,569 | 4559 | 16,128 |
| | | % | 43.86% | 78.28% | 50.09% |
| | | Yes | 14,806 | 1265 | 16,071 |
| | | % | 56.14% | 21.72% | 49.91% |
| | | Total | 26,375 | 5824 | 32,199 |
| | | % | 81.91% | 18.09% | 100.00% |

The behavior of owning a property title, the area of residence, and membership in an association also additionally explained the relationship of household public services with poverty. On average, the population who held the ownership title to their properties reached 38.48%. With respect to belonging to a political or neighborhood association, peasant rounds, irrigators, professional associations, or agricultural association, among others, the respondents indicated that on average only 21.87% belonged to any of the aforementioned. In the case of the area of residence, on average 22.24% of the members of the households lived in rural areas (Table 2). The relationship between holding the title of the property, belonging to an association or organization, and having a rural residence area with poverty indicated that the populations in a non-poor condition who did not have a property title, were not part of an association, and did not have a residence in a rural area reached 60.43%, 78.10%, and 68.89%, respectively, while those in a non-poor condition who did have a property title, were part of an association, and had a residence in a rural area reached 39.57%, 21.90%, and 31.11%. On the contrary, the populations in a poor condition who did not have a property title, were not part of an association, and did not have a residence in a rural area reached 79.46%, 82.11%, and 43.06%, and those in a poor condition who had a property title, were part of an association, and had a residence in a rural area reached 36.13%, 21.17%, and 35.78%, respectively (Table 6).

After analyzing the behavior of household public services, their relationship with poverty was analyzed. In the case of basic public services in the household, potable water, sanitation, electricity, and cell phone services had negative relationships with poverty, given that the Pearson's p-values were −0.0970, −0.2015, −0.1313, and −0.0817, determining that there were low negative correlations (Table 7).

Analyzing the correlation between the characteristics of the household members with poverty, it could be seen that the level of higher, secondary, and primary education and receiving remittances had inverse relationships with poverty, given that the Pearson's p-values were equal to −0.1678, −0.1784, −0.1190, and −0.0531, indicating that they had low negative correlations. However, in the case of the number of household members, it had a positive relationship with poverty, given that it had a Pearson p-value equal to 0.2591 (Table 8).

**Table 6.** Relationship between household belonging and area of residence with poverty.

| Item | Type of Service | Poverty Value | Not | Yes | Total |
|---|---|---|---|---|---|
| Belonging, area, and associations of the household | Has title deed | Not | 15,938 | 4628 | 20,566 |
| | | % | 60.43% | 79.46% | 63.87% |
| | | Yes | 10,437 | 1196 | 11,633 |
| | | % | 39.57% | 20.54% | 36.13% |
| | | Total | 26,375 | 5824 | 32,199 |
| | | % | 81.91% | 18.09% | 100.00% |
| | Is part of a partnership | Not | 20,599 | 4782 | 25,381 |
| | | % | 78.10% | 82.11% | 78.83% |
| | | Yes | 5776 | 1042 | 6818 |
| | | % | 21.90% | 17.89% | 21.17% |
| | | Total | 26,375 | 5824 | 32,199 |
| | | % | 81.91% | 18.09% | 100.00% |
| | Rural area of residence | Not | 18,170 | 2508 | 20,678 |
| | | % | 68.89% | 43.06% | 64.22% |
| | | Yes | 8205 | 3316 | 11,521 |
| | | % | 31.11% | 56.94% | 35.78% |
| | | Total | 26,375 | 5824 | 32,199 |
| | | % | 81.91% | 18.09% | 100.00% |

**Table 7.** Correlation between basic household public services with poverty.

| Variable | Poverty | Water Service | Sanitation Service | Electric Energy Service | Cell Phone Service |
|---|---|---|---|---|---|
| Poverty | 1.0000 | | | | |
| Water service | −0.0970 | 1.0000 | | | |
| Sanitation service | −0.2015 | 0.5219 | 1.0000 | | |
| Electric energy service | −0.1313 | 0.3150 | 0.3161 | 1.0000 | |
| Cell phone service | −0.0817 | 0.1229 | 0.1823 | 0.2453 | 1.0000 |

**Table 8.** Correlation between household member characteristics with poverty.

| Variable | Poverty | Higher Education Level | Secondary Education Level | Primary Education Level | Household Members | It Has Recall |
|---|---|---|---|---|---|---|
| Poverty | 1.0000 | | | | | |
| Higher education level | −0.1678 | 1.0000 | | | | |
| Secondary education level | −0.1784 | 0.4724 | 1.0000 | | | |
| Primary education level | −0.1190 | 0.2573 | 0.5446 | 1.0000 | | |
| Household members | 0.2591 | −0.0268 | 0.0465 | 0.1316 | 1.0000 | |
| Has remittance | −0.0531 | −0.0609 | −0.0984 | −0.1057 | −0.1768 | 1.0000 |

In the analysis of the correlation between the complementary assets and services of the household with poverty, it was determined that household complementary assets and services such as having a washing machine, motorcycle, tricycle, motorcycle taxi, computer, stove, and refrigerator had inverse or negative correlations with poverty; since the Pearson's p-values were equal to −0.2128, −0.0438, −0.0012, −0.0322, −0.2255, −0.2025, and −0.2649, indicating that they had low negative correlations (Table 9).

**Table 9.** Correlation between complementary household assets and services with poverty.

| Variable | Poverty | It Has Washing Machine | It Has a Motorcycle | It Has Tricycle | It Has Mototaxi | It Has a Computer | It Has Kitchen | It Has Refrigerator |
|---|---|---|---|---|---|---|---|---|
| Poverty | 1.0000 | | | | | | | |
| Has a washing machine | −0.2128 | 1.0000 | | | | | | |
| Has a motorcycle | −0.0438 | 0.0585 | 1.0000 | | | | | |
| Has a tricycle | −0.0012 | −0.0246 | 0.0064 | 1.0000 | | | | |
| Has a motorcyle taxi | −0.0322 | −0.0079 | 0.0533 | −0.0064 | 1.0000 | | | |
| Has a computer | −0.2255 | 0.4835 | 0.0927 | −0.0059 | −0.0072 | 1.0000 | | |
| Has a kitchen | −0.2025 | 0.2106 | 0.0786 | 0.0266 | 0.0579 | 0.2254 | 1.0000 | |
| Has a refrigerator | −0.2649 | 0.5111 | 0.0642 | −0.0231 | 0.0801 | 0.4243 | 0.3277 | 1.0000 |

With respect to the correlations between belonging, citizen participation, and area of residence with poverty, the variables such as having a property title and being part of a civil, political, cultural, religious, or other association or organization had negative relationships with poverty. It was determined that there were low inverse correlations, given that the Pearson's $p$-values were equal to −0.1525 and −0.0378. On the contrary, there was a low positive correlation between having a rural residence area and poverty, given that the Pearson's $p$-value was equal to 0.2074 (Table 10).

**Table 10.** Correlation between household belonging and area of residence with poverty.

| Variable | Poverty | Has Title Deed | It Is Part of a Partnership | Rural Area of Residence |
|---|---|---|---|---|
| Poverty | 1.0000 | | | |
| Has title deed | −0.1525 | 1.0000 | | |
| It is part of a partnership | −0.0378 | 0.0152 | 1.0000 | |
| Rural area of residence | 0.2074 | −0.3465 | −0.0025 | 1.0000 |

After analyzing the behavior of the variables that explained the effects of public services in the household on poverty, regression analysis was carried out applying the logit-binomial model using the statistical program Stata 16.0. The results are shown in Table 11. All of the independent variables had individual significance at the 95% confidence level, and the Z-values were greater than 2 in absolute value or had a P > |z| less than 0.05. In addition, based on having a Prob value > chi2 equal to 0.0000 and less than 0.05, it could be said that they had global significance; therefore, they met the conditions of having global and individual significance. Complementarily, when performing an analysis of the other statistics, a pseudo R2 value of 21.89% and a log pseudo likelihood value of −3,546,765.6 were obtained, which showed that the model was consistent and efficient. Therefore, its analysis and interpretation were appropriate to explain the effects of public services on household poverty in Peru (Table 11).

Analyzing the coefficients and contrasting with economic theory, it could be determined that access to potable water, sanitation, electricity, and cell phone services; the level of higher, secondary, and primary education; having a washing machine, motorcycle, tricycle, motorcycle taxi, kitchen, computer, and refrigerator; having a property title; being part of an association; a rural residence area; and having a remittance had negative effects on poverty in households in Peru. However, the number of household members had a positive effect on poverty (Table 11).

The values obtained for "Exp (Coefficient)" that showed the OR (odds ratio) represented the division between the probability that the event that defined the dependent variable would occur versus the probability that it would not occur in the presence or absence of the factor. In the case of access to basic household services such as potable water, sanitation, electricity, and cell phone services, the values were 0.9763, 0.7283, 0.9446, and 0.6544. These values indicated that, among those who had access to potable water, sanitation, electricity and cell phone services, the coefficients between the probability of being poor were 0.9763, 0.7283, 0.9446, and 0.6544 times larger, than those of not being

poor among those who did not have access to potable water, sanitation, electricity, and cell phone services (Table 11).

**Table 11.** Regression results of household public services with poverty.

| Variable | Coefficient | Standard Error | Z-Value | P > \|z\| | [95% Conf. Interval] | | Exp (Coef.) | Marginal Effects |
|---|---|---|---|---|---|---|---|---|
| Water service | −0.024 | 0.003 | −8.630 | 0.000 | −0.030 | −0.019 | 0.9763 | −0.003 |
| Sanitation service | −0.317 | 0.003 | −116.850 | 0.000 | −0.323 | −0.312 | 0.7283 | −0.038 |
| Electric energy service | −0.057 | 0.004 | −13.590 | 0.000 | −0.065 | −0.048 | 0.9446 | −0.007 |
| Cell phone service | −0.424 | 0.004 | −106.450 | 0.000 | −0.432 | −0.416 | 0.6544 | −0.056 |
| Higher education level | −0.443 | 0.004 | −109.210 | 0.000 | −0.451 | −0.435 | 0.6421 | −0.046 |
| Secondary education level | −0.147 | 0.002 | −60.480 | 0.000 | −0.151 | −0.142 | 0.8633 | −0.017 |
| Primary education level | −0.151 | 0.003 | −59.650 | 0.000 | −0.156 | −0.146 | 0.8598 | −0.018 |
| Has a washing machine | −0.486 | 0.003 | −158.440 | 0.000 | −0.492 | −0.480 | 0.6151 | −0.052 |
| Has a motorcycle | −0.473 | 0.003 | −147.330 | 0.000 | −0.480 | −0.467 | 0.6231 | −0.047 |
| Has a tricycle | −0.021 | 0.008 | −2.610 | 0.009 | −0.005 | −0.037 | 0.9792 | −0.002 |
| Has a motorcycle taxi | −0.600 | 0.004 | −154.050 | 0.000 | −0.607 | −0.592 | 0.5488 | −0.057 |
| Has a computer | −1.080 | 0.003 | −381.130 | 0.000 | −1.086 | −1.075 | 0.3396 | −0.111 |
| Has a kitchen | −0.506 | 0.003 | −170.760 | 0.000 | −0.512 | −0.500 | 0.6029 | −0.067 |
| Has a refrigerator | −0.739 | 0.002 | −310.370 | 0.000 | −0.744 | −0.735 | 0.4776 | −0.088 |
| Has a property title | −0.373 | 0.002 | −163.680 | 0.000 | −0.377 | −0.369 | 0.6887 | −0.041 |
| Is part of an association | −0.213 | 0.002 | −88.210 | 0.000 | −0.217 | −0.208 | 0.8082 | −0.023 |
| Household members | 0.566 | 0.001 | 926.500 | 0.000 | 0.564 | 0.567 | 1.7612 | 0.065 |
| Rural residence zone | −0.223 | 0.003 | −81.970 | 0.000 | −0.228 | −0.218 | 0.8001 | −0.024 |
| Has remittance | −0.188 | 0.004 | −51.090 | 0.000 | −0.195 | −0.181 | 0.8286 | −0.020 |
| Constant | −1.108 | 0.005 | −202.390 | 0.000 | −1.119 | −1.098 | 0.3302 | − |
| Logistic regression model | | | Number of observations | | = | 32,199 | | |
| | | | LR chi2(19) | | = | 1,988,013.630 | | |
| | | | Prob > chi2 | | = | 0.0000 | | |
| Log likelihood | −3546765.6 | | Pseudo R2 | | = | 0.2189 | | |
| | | | y | | = | Pr(Poor) (predict) | | |
| | | | | | = | 0.1320 | | |

In addition, when considering the characteristics of the household members, such as higher, secondary, and primary education levels, the number of household members, and receiving remittance, the values of "Exp (Coefficient)" were 0.6421, 0.8633, 0.8598, 1.7612, and 0.8286. These values showed that, among those who reached the level of higher, secondary, and primary education, had fewer members of the household, and had remittances, the coefficients between the probability of being poor were 0.6421, 0.8633, 0.8598, 1.7612, and 0.8286 times larger, respectfully, than the coefficients between the probability of not being poor among those who did not reach the level of higher, secondary, and primary education, had many members of the household, and did not have a remittance (Table 11). Therefore, the number of household members was a risk factor for increasing poverty.

When considering the complementary assets and services of the household, such as having a motorcycle, tricycle, motorcycle taxi, computer, kitchen, refrigerator, and washing machine, the values of "Exp (Coefficient)" were 0.6151, 0.6231, 0.9792, 0.5488, 0.3396, 0.6029, and 0.4776. These values showed that, among those who had a motorcycle, tricycle, motorcycle taxi, computer, kitchen, refrigerator, and washing machine at the household; the coefficients between the probability of being poor were higher than those of not being poor among those who did not have a motorcycle, tricycle, mototaxi, computer, kitchen, refrigerator, and washing machine at the household (Table 11).

In the case of belonging and area of residence of the household, the values of "Exp (Coefficient)" for having a property title, being part of an association, and having a rural residence area were 0.6887, 0.8082, and 0.8001. These values showed that, among those who had a property title, were part of an association, and had a rural area of residence, the coefficients between the probability of being poor were 0.6887, 0.8082, and 0.8001 times larger, than those of not being poor among those who did not have a property title, were not part of an association, and had an urban area of residence (Table 11).

Next, when carrying out the analysis of the marginal effects of the established model, access to potable water, sanitation, electricity, and cell phone services in the household had negative effects on poverty, given that an increase in the provision of potable water, sanitation, electricity, and cell phone services in the household decreased the probability of being poor by 0.3, 3.8, 0.7, and 5.6 percentage units. In the case of the level of higher, secondary and primary education and having remittances, these had negative effects on poverty and the number of household members had a positive effect on poverty, since an increase in the level of higher education, secondary; and primary education and receiving a remittance decreased the probability of being poor in 4.6, 1.7 and 1.8 percentage units. Meanwhile, if the number of members of the household increased by one person, then the probability of being poor increased by 6.5 percentage units (Table 11).

Regarding complementary assets and services in the household, the variables of having a washing machine, motorcycle, tricycle, motorcycle taxi, computer, kitchen, and refrigerator had negative effects on poverty in households in Peru, given that if the household had a washing machine, motorcycle, tricycle, motorcycle taxi, computer, kitchen, and refrigerator, then the probability of being poor decreased by 5.2, 4.7, 0.2, 5.7, 11.10, 6.7, and 8.80 percentage units.

Finally, having a property title, being part of an association, and a rural residence area had negative effects on poverty, given that upon obtaining a property title, forming part of an association, and having a residence in a rural area, then the probability of being poor decreased by 4.1, 2.3, and 2.4 percentage units.

## 5. Discussion

After having obtained the expected results and analyzing the behavior between the variables that influenced and had an effect on poverty, we verified the research hypothesis that households that have greater access to residential public services are less likely to be poor.

As a theoretical basis for discussion, poverty generated by exclusion from access to public services is considered, the phenomenon of which is caused by the deprivation of basic public services in the household, especially education, health, and the provision of potable water, sanitation, electricity, and telephone services, where said denial of access became the source of poverty and extreme poverty. This is complemented by limitations in the growth of economic income due to the lack of access to capital, land, and adequate, quality, and efficient public services (Barillas 2010; Verdera 2007). In this sense, the solution to reducing poverty is through equitable public policies, which must include the sufficient provision of public services and infrastructure, making it an economic, social, and political agenda that must be considered at all levels of government (Dammert Lira and García Carpio 2011).

In this sense, as in the present investigation, the effects of access to potable water, sanitation, and electricity services on poverty in Peru coincide with what was found by Vargas (2012), since this author identified the existence of a negative relationship between public infrastructure services and poverty and, at the same time, demonstrated that the poorest regions had less access to potable water, sewage, and electricity services. In addition, the impact that access to water, sanitation, and electricity services had on poverty at the regional level in Peru was decisive, given that the poorest regions had lower levels of access to water, drainage, and electricity, an issue that was also evidenced in our research. In this sense, based on these results, the inclusion of some public policies that guarantee greater access to public household services could be considered, given that this will reduce the inequality and inequity gap that exists between different regions of Peru and the social and economic gaps that still persist in the geographical areas.

The proposition made by Aparicio et al. (2011), who, like this research, considered telephone, electricity, water, and drainage to be part of public services and their infrastructure and part of the physical assets that guarantee the functioning of the household, demonstrated that access to different types of public infrastructure allowed improvement

of quality of life and therefore reduced the conditions of poverty. What stands out is that the effects of infrastructure on poverty are long-term. When applying a logit-binomial econometric model in said research, access to the telephone was the infrastructure with the greatest impact, but in our case, the one that had the greatest effect was access to a computer.

In addition to the present investigation, the results obtained coincide with what was determined by Cuenca-López and Torres (2020), since by applying a panel model these authors were able to demonstrate that access to economic infrastructure such as transportation and electricity in households had a negative effect on both levels of poverty. In addition, access to social infrastructure such as education significantly decreased poverty. In this sense, this study demonstrates in the same way as those previously indicated that the establishment of public policies for the reduction of poverty must be oriented through the actions of social and economic projects and programs, seeking to cover quality services with full access. In addition, consistent with what was determined by Masika and Baden (1997) via the application of a logistic model, establishing access to public telephone service, electricity, and sanitation in the household reduces the probability that the household will be poor.

Finally, the findings of the study agree with what was determined by Parra (2011) in Colombia, who established that access to public services, use of adequate targeting, consideration of the subsidy rate, and considering the amount consumed among the beneficiaries of the subsidies showed great relevance for poverty. In the case of poor households, access to services is less, with the exception of electricity service, in which case coverage is practically universal. Therefore, covering these gaps is a universal necessity and guaranteeing the reduction of said gaps will allow the expected well-being of the population to be achieved.

## 6. Conclusions

According to the results obtained based on the existing information in the ENAHO database of the INEI for 2021, poverty on average reached 19.63%, access to potable water service on average was 82.80%, access to sanitation service on average was 66.42%, access to electricity service on average was 95.52%, access to cell phone service on average was 94.53%, those who reached the level of higher education on average was 16.62%, achievement of a secondary education level on average was 51.15%, reaching the primary education level on average was 78.09%, the number of household members on average was 3 people per household, and only 8.97% indicated that they had received remittances. On average, 29.74%, 11.13%, 1.22%, 6.82%, 33.94%, 89.71%, and 55.05% of households had a washing machine, motorcycle, tricycle, motorcycle taxi, computer, kitchen, and refrigerator, respectively. In the case of holding a property title, on average this reached only 38.48%, belonging to an association only reached 21.87%, and those residing in rural areas reached only 22.24% of those surveyed.

Finally, it was determined that access to potable water, sanitation, electricity, and cell phone services; the level of higher, secondary, and primary education; having a washing machine, motorcycle, tricycle, motorcycle taxi, computer, kitchen, and refrigerator in the household; having a property title; being part of an association; having a rural residence area; and having remittances had negative effects on poverty in households in Peru. However, the number of household members had a positive effect on poverty.

## 7. Limitations and Future Recommendations

Full access to statistical data was the very determining limitation in this research. There weren't many precedents linking household utilities to poverty; given that, given that there was a diversity of methodologies, research should continue to be developed that demonstrates the behavior of these two variables with cross-sectional information, panel data, or time series.

It is recommended that political decision-makers consider this study as a starting point for the design of public policies that allow reducing poverty in Peru, in view of the fact that it is not enough to propose legal norms that limit the normal development of families and households in society.

The study can be used as a starting point for research exploring public services as they relate to economic income, since this can show the other side of the coin.

**Author Contributions:** The authors contributed to this research according to the following details: Conceptualization, S.L.A.-P.; methodology, D.A.C.-Á.; software, R.M.-P.; validation, M.F.L.-Á.; formal analysis, N.J.U.-G. and A.M.-F.; data curation, B.E.C.-Q.; research, writing—preparing the original draft, writing—reviewing and editing, J.C.Q.-M. and M.Q.-L.; display, D.J.I.-L.; supervision, R.N.T.-H. All authors have read and agreed to the published version of the manuscript.

**Funding:** This research received no external funding.

**Data Availability Statement:** The data used in this research are available to the public in the databases of the National Household Survey (ENAHO) and the National Institute of Statistics and Informatics (INEI) of Peru.

**Conflicts of Interest:** The authors declare that they have no conflict of interest of any kind in this scientific research.

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
