# Peer review of "Public Services in the Household and Their Effect on Poverty, Analysis for the Peruvian Case, 2021"

_socsci, doi:10.3390/socsci12060328_

Round 1

Reviewer 1 Report

I reviewed the draft with great interest entitled “Public services in the household and their effect on poverty, analysis for the Peruvian case, 2021”. The paper is dealing with an interesting side of the research. However, I have a few comments for the authors:

1.       The introduction section is very long and the study is lacking literature as well as theoretical discussion. Thus, I suggest restructuring this section and splitting it into an introduction and literature review section. The introduction section should only focus to discuss the factors affecting household poverty using in the current study. All the literature evidence and discussion should move into the literature section.

2.       What do you mean by this statement “household on poverty in Peru”? Is this household poverty in Peru? The complete draft needs careful proofreading to avoid such mistakes.

3.       Which underpinning theory supports the current model? Authors must develop the Hypothesis development section separately. The hypotheses should be developed with the support of relevant theory (s). The following articles may help as these articles are directly related to the current study. shorturl.at/cfqG4; 10.1007/s11135-018-0710-0; 10.1111/aswp.12152; 10.1007/s11135-021-01205-8; 10.1109/DASA54658.2022.9765280

4.       Remove the unnecessary subheadings from the section Materials and Methods.

5.       What are the operational definitions of the variables? How are these variables measured in the current study?

6.       What is the poverty line used for this study? Discuss this under the econometric equation.  

7.       Table 10, Regression findings, the authors analyzed the coefficient of each variable which they obtained from logit-binomial regression. However, when the logit-binomial model is used for analysis, the author must calculate the ODD ratios and then must interpret those odd ratios instead of analyzing the coefficient values directly. Revised these results and their interpretations accordingly.

8.       The authors only interpret the current findings, there is no discussion of findings with valid arguments and also the current findings have not been aligned with existing literature.

9.       What are the practical and theoretical implications of the study? It is advised to add this section before the conclusion section.

10.   Also, add the limitations and future recommendations in a separate section 

Author Response

Good evening.

I must thank you for the comments made to the scientific article that we worked on as a team.

Everything possible was done to lift the observations made, the final result of which is attached to this, where the final file of the scientific article was considerably improved.

We hope we have been able to work in an orderly and coherent manner.

We thank you for your contributions.

Reviewer 2 Report

The paper approaches an important topic related to the provision of public services and poverty in Peru after the pandemic in 2021.

I will comment on some issues that I feel are important to take into account in order to improve this article:

Methodological issues:

1) there is no clear justification of the inclusion of some of the variables in the study that were taken from the database of the INEI, e.g. citizenship participation... why did the authors decide to include citizenship participation (described as "is part of an association" or "is part of partnership") and also "title of property" as part of the group of variables analized as public services? when being part of an association can not be considered as public service provided by the state. And evene more, when the purpose of the study is to establish the relationship that exists with the effects of public services and poverty. So, how does this variable fit in the study? and the same applies to property title...

2) Furthermore, there are no references to studies that have analysed citizen participation and its relationship with access to public services and poverty.

3) Table 5 should include in the title the variable studied "citizenship participation" (it is part of a partnership)... but how is this variable being measured? whose participation is taking into account? participation of all members of the household? of only one member participates it counts as such? and participation in what type of associations?

Also, this variable of participation is included in table 5 as type of service, when it should not be considered as such if we consider that being a member of an association can not be be considered as a public service provided by the state. and the same happens with rest of variables included in this table: area of residence and having a property title. 

4) The results in Table 5 described in the text are sometimes difficult to understand, so it would be advisable to rewrite this part of the text (lines 304-314).

5) Table 9 should also include in the title the variable studied "citizenship participation" (it is part of a partnership) as well as the other 2 variables included in this table. Besides, there is no reference to this variable in the description of results obtanied when analysing the correlation of the three variables included in table 9. It is important to describe these results in relation to participation issues. 

6) with respect to the variable "area of rural residence", this variable could not be dichotomous (0,1), but have more scores (e.g. 0,1,2) that could better describe the inequalities found in Peru with respect access to public services by area of household residence. That is, how access to services varies between rural and urban areas and, within urban areas, by city size.

Conclusions:

7) More work needs to be done on the conclusions. The conclusions are not well elaborated or developed. In general, they are limited to describe again the statistics of public services that have already been described previously in table 1, and which form part of the information of the database of the INEI for 2021.

8) There is no reference to the level of impact or influence that the lack of access to public services has in people´s quality of life in the different regions of Peru analyzed in this study. Nor the importance of promoting specific policy actions that can improve and promote the development of people in a more vulnerable or disadvantaged position and in the poverty situation. 

9) Along with this, there are any references to the limitations of the study or possible lines for future research. 

Author Response

Good evening.

I must tell you that his observations were pertinent and were considered as far as possible in the scientific article, so we reach the corrected version, where what was considered by his person was incorporated, so that it can be validated by the person of he.

We will always be very grateful for his contribution to this investigation.

Greetings.

Round 2

Reviewer 1 Report

Authors followed the reviewers’ comments, and the revised draft is more reader friendly. I suggest accepting the current draft.